# NAFLD Preclinical Models: More than a Handful, Less of a Concern?

**DOI:** 10.3390/biomedicines8020028

**Published:** 2020-02-08

**Authors:** Yvonne Oligschlaeger, Ronit Shiri-Sverdlov

**Affiliations:** Department of Molecular Genetics, School of Nutrition and Translational Research in Metabolism (NUTRIM), Maastricht University, Universiteitssingel 50, 6229 ER, Maastricht, The Netherlands; r.sverdlov@maastrichtuniversity.nl

**Keywords:** NAFLD, mouse models, multifactorial disease, translational value

## Abstract

Non-alcoholic fatty liver disease (NAFLD) is a spectrum of liver diseases ranging from simple steatosis to non-alcoholic steatohepatitis, fibrosis, cirrhosis, and/or hepatocellular carcinoma. Due to its increasing prevalence, NAFLD is currently a major public health concern. Although a wide variety of preclinical models have contributed to better understanding the pathophysiology of NAFLD, it is not always obvious which model is best suitable for addressing a specific research question. This review provides insights into currently existing models, mainly focusing on murine models, which is of great importance to aid in the identification of novel therapeutic options for human NAFLD.

## 1. Introduction 

Due to excess intake of fat- and/or sugar-enriched diets and a lack of exercise, overweight and obesity are currently a major and continuously growing public health concern. Strongly associated with the increasing trend in obesity is the metabolic syndrome (MetS), in which disturbed lipid homeostasis and metabolic inflammation are taking the lead. Besides type 2 diabetes (T2D) and cardiovascular diseases, MetS increases the risk of developing non-alcoholic fatty liver disease (NAFLD), the most prevalent chronic liver disease worldwide [1]. The spectrum of NAFLD ranges from benign simple steatosis [2] to steatohepatitis (NASH) and end-stage liver diseases. If steatosis is not managed in time, resident liver cells (e.g., Kupffer and hepatic stellate cells) become activated and immune cells (mainly macrophages) infiltrate the liver, a condition defined as NASH. This progressive form of NAFLD [3] can further trigger hepatocyte damage with/without fibrosis and increase the risk of cirrhosis [4] and hepatocellular carcinoma (HCC) [5]. In contrast to alcoholic liver disease- and viral hepatitis-induced HCC, NASH-related HCC is currently the most rapid growing indication for liver transplant in HCC patients [6]. 

Obviously, there is an urgent need for a reliable ‘humanized’ model that displays a liver phenotype that is identical to human disease, with macrovesicular steatosis, lobular inflammation, hepatocellular ballooning (including Mallory-Denk bodies), and fibrosis as predominant features. Subsequently, an optimal NAFLD model should have the ability to further progress to advanced fibrosis, cirrhosis, and ultimately HCC. Moreover, it should encompass MetS-related characteristics, such as obesity, disturbed lipid, glucose and insulin metabolism, as well as systemic inflammation. So far, using a wide variety of preclinical models, considerable efforts have recently been made to better understand the pathogenesis of human NAFLD and/or related clinical questions. However, none of these models resemble the complete human NAFLD spectrum, including related metabolic features that recapitulate this chronic liver disease. For instance, several models, exposed to various dietary compositions, have broadened our knowledge with regard to NAFLD progression, in particular early-onset low-grade inflammation. Alternative models (genetic or chemically-induced) provided insights into fibrotic features of human NAFLD, one of the most important predictors of human NASH progression [7]. Other models have been more suitable for testing therapeutic interventions in the context of NAFLD/NASH. Nevertheless, there are still significant unmet needs with regard to non-invasive diagnostic methods, therapeutic target identification, and drug development, which implies the need for robust preclinical models [8]. In the current Review, we will provide an update on existing preclinical NAFLD models. This Review focuses on rodents, mainly on mouse models, which are relatively low in costs and therefore allow for studying metabolic and genetic drivers of NAFLD/NASH within a considerable time frame. Moreover, these models serve as an important and well-controlled tool for preclinical drug testing in a multisystemic environment.

## 2. Insights into Available Preclinical Models for Non-alcoholic Fatty Liver Disease

A wide variety of dietary, genetic, chemically-induced, and/or other rodent models ([8,9,10,11,12,13,14,15,16]) have greatly advanced our understandings on NAFLD pathophysiology, as will be discussed in the following sections (see also Figure 1, Table 1). 

### 2.1. Dietary Murine Models

Diet-induced obesity is known to be the most common risk factor for NAFLD in humans [17]. The experimental NAFLD/NASH models are often based on overnutrition, a condition that can be induced by means of diets varying in macronutrient composition, amongst others.

One of the simplest ways of disturbing lipid metabolism, inducing steatosis and moderate NASH, is by the administration of a regular high-fat diet (HFD; 60% fat, 20% proteins, 20% carbohydrates) [18,19]. For instance, feeding wild-type C57BL/6 mice a HFD for 10–12 weeks resulted in phenotypic changes such as hyperlipidemia, hyperinsulinemia, and glucose intolerance [19,20]. Despite hepatic steatosis, inflammatory cell infiltration was not present until 19 weeks of HFD intake [19]. While after long-term (34–36 weeks) HFD feeding, significant increases in circulating liver enzyme levels, i.e., alanine aminotransferase (ALT) and aspartate aminotransferase (AST) were observed [19], these mice showed only minor signs of inflammation and fibrosis [21], even after prolonged administration up to 50 weeks [19]. Yet, after chronic feeding (80 weeks) of HFD, which mimics lifetime HFD consumption and enables proper design of treatment options, Velázquez et al. [17] demonstrated that mice displayed obesity and insulin resistance. In addition, these mice were shown to develop NAFLD features, including hepatic steatosis, cell injury, portal and lobular inflammation, hepatic ER stress, as well as fibrosis [17]. In line with Chen et al. [22], which showed a deterioration in NAFLD when germ-free mice were inoculated with the Firmicutes phyla, they also found increases in the Firmicutes phyla in response to prolonged HFD [17]. Though intestinal permeability was not measured in this study, these data pointed towards diet-induced gut-microbial dysbiosis [17], a well-known microbial event that has been previously observed in NAFLD patients [23]. Yet, it is unclear whether a similar but shorter dietary intervention would also provide insights into NAFLD progression [17].

In contrast to regular HFD, the atherogenic diet, composed of 1.25% cholesterol plus 0.5% cholate, resulted in increased plasma and liver lipid levels and was shown to induce NASH with hepatocellular ballooning in a time-dependent manner from 6–24 weeks [24]. Notably, the addition of a high-fat component exacerbated the histologic severity of NASH, and resulted in hepatic insulin resistance, oxidative stress, and activation of hepatic stellate cells [24]. Exposure to HFD containing 0.1–2.0% cholesterol (HFC) [25,26] for seven months in murine models, such as wild-type C57BL/6 mice, resulted in the development of obesity, hepatomegaly, hepatic steatosis, and varying degrees of steatohepatitis [27]. 

Due to the variability in disease onset and limited development of fibrosis, a novel so-called Amylin Liver NASH (AMLN) model was generated [28]. This model covers a high-fat/high-fructose (40%/22%) diet containing trans-fatty acids (~18%) and high-cholesterol (2%), thereby better resembling the Western-type diet and subsequent development of NASH features. Remarkably, only after 26–30 weeks of feeding an AMLN diet, wild-type C57BL/6 developed marked steatosis, moderate lobular inflammation and hepatocellular ballooning [29]. However, when obese leptin-deficient *ob/ob* mice were fed a similar AMLN diet for 12 weeks, mice displayed an accelerated and more pronounced metabolic NASH phenotype as compared to wild-type C57BL/6 [29]. Indeed, it is well-known that *ob/ob* mice, which carry a homozygous mutation in the leptin gene that protect it from binding to its receptor, are susceptible to insulin resistance and T2D, thus being predisposed to metabolic features resembling NAFLD [30]. Yet, spontaneous progression from simple steatosis to NASH and hepatic fibrosis is rather prevented in these mice [31], pointing towards the need of a second stimulus. 

More recently, according to the FDA-ban on trans-fats as food additives [32], another obesogenic trans-fat-free diet substituted with saturated fat (palm oil) was explored [33]. This so-called Gubra Amylin NASH (GAN) diet has a nutrient composition and caloric density (40% high-fat, 22% high-fructose 2% high-cholesterol) similar to AMLN diet. Upon feeding *ob/ob* mice GAN diet for 16 weeks, animals displayed biopsy-confirmed liver lesions with features of fibrotic NASH. While these features were similar to AMLN-fed *ob/ob* mice, GAN-fed *ob/ob* mice showed a more pronounced weight gain and increased adiposity. In contrast, wild-type C57BL/6 mice required a prolonged feeding period (28 weeks) of GAN diet to induce consistent fibrotic NASH. However, compared to AMLN diet, GAN-fed wild-type mice had significantly greater body weight gain. Altogether, obesogenic GAN diet induces hallmarks of fibrotic NASH in both models [33], suggesting its suitability for preclinical therapeutic testing against NASH. 

Administering an alternative fast-food-like nutritional regime based on high-fat/high-fructose/high-cholesterol (41%/30%/2%) was also shown to induce NASH in various genotypes [34]. These models included wild-type C57BL/6, *ob/ob* mice as well as KK-A^y^ [35] mice, the latter carrying a mutation in the Agouti gene that increases its susceptibility to human NAFLD-like metabolic alterations [36]. Relevantly, Abe et al. [34] showed that *ob/ob* mice under these conditions displayed more pronounced NAFLD activity score, fibrosis progression, obesity and hyperinsulinemia compared to the other models. Given that the metabolic, histologic, and transcriptomic features observed in *ob/ob* mice were similar to human NASH, this model may be further explored as a potential preclinical tool to discover novel drugs for NASH [34].

Relevantly, Henkel et al. [37] explored the impact of long-term exposure (20 weeks) with a high-caloric (43%) Western-type diet composed of soy-bean oil (high n-6-PUFA, 25g/100g) and 0.75% cholesterol. In contrast to cholesterol-free HFD [38], dietary cholesterol in soybean oil resulted in increased Kupffer cell activation and oxidative stress as well as hepatic steatosis, ballooning, inflammation and fibrosis in wild-type C57BL/6 [37], which closely resembles clinical NASH features. In line, when mice were fed an alternative high-caloric (45%) cholesterol-free HFD (composed of lard (21g/100g)/soy-bean oil (3g/100g)/5% fructose in drinking water), only mild steatosis and no signs of hepatic inflammation and fibrosis were observed [37]. Thus, in agreement with previous studies [25,26,38,39,40], these findings indicate that the supplementation of dietary cholesterol triggers experimental hepatic inflammation and fibrosis [37]. 

Other dietary variants were explored by Montandon et al. [41], comparing the high-fat atherogenic diet (60% fat plus 1.25% cholesterol and 0.5% cholic acid) versus the commonly used methionine/choline-deficient diet (MCD). In line with others [24,42], wild-type C57BL/6 mice fed a cholesterol/cholate-rich diet showed increases in hepatic cholesterol and free fatty acids, while MCD mice predominantly accumulated triglycerides in their livers [41]. Strikingly, MCD caused a reduction in liver weights, whereas atherogenic diet did not [41]. Moreover, MCD increased hepatic damage, lobular inflammation, lipogranulomas, tissue fibrosis, and liver enzymes compared to mice fed a cholesterol/cholate-rich diet. In addition, transcriptional analyses revealed a dysregulation in extracellular matrix remodeling and hepatic stellate cell activation in response to MCD, but not an atherogenic diet [41]. Altogether, these data pointed towards a more severe form of NASH in MCD mice [41], which was in line with previous studies showing that MCD triggered extensive hepatic inflammation in rats [43] and mice [44,45,46] within a very short time frame. To overcome the lack of severe hepatic fibrosis often observed in preclinical models, mice are commonly fed MCD [47] or a diet low in/deficient for choline (CD) [48]. Although CD exacerbated fatty liver [48], MCD resulted in rapid NASH development with severe liver fibrosis within 4–10 weeks, likely as a consequence of reduced VLDL synthesis and hepatic-β oxidation [20]. While liver inflammation and elevated circulating liver enzyme levels returned back to normal levels after switching the diet back to control within 16 weeks, fibrosis and CD68-positive macrophages remained present [47]. 

It is also interesting to note that leptin-resistant *db/db* mice (which carry a mutation in the leptin receptor gene [49] and lack the ability to spontaneously develop hepatic inflammation [35,50]) displayed marked hepatic inflammation and fibrosis in response to feeding an MCD diet for four weeks [45]. These data suggest that, similar to *ob/ob* mice [31,51], *db/db* mice need a second stimulus to induce NASH [45]. Nevertheless, it is noteworthy that all MCD models rather showed significant reductions in weight, concomitant loss in liver mass and cachexia, as well as low serum levels of insulin, fasting glucose, leptin and triglycerides, and a lack of insulin resistance [12,45,52]. Given that these preclinical observations are opposite to the effects seen in overweight and obese individuals with NAFLD, these data suggest that the use of MCD models as preclinical tools to represent human NAFLD is rather limited [53]. Further, though *ob/ob* and *db/db* models serve as useful preclinical tools that mimic insulin resistance as observed in humans, it should be kept in mind that these mice bear mutations that are not prevalent in obese humans or NASH patients.

Relevantly, compared to the MCD diet, mice on a choline-deficient L-amino acid-defined (CDAA) diet developed a more severe degree of NASH and fibrosis, while not having any signs of weight loss [54]. Yet, only after long-term feeding, i.e., 5–6 months with CDAA diet, wild-type C57BL/6 mice displayed increased plasma lipid levels and HOMA-IR, pointing towards the development of insulin resistance [55]. It is relevant to note that the combination of CDAA with HFD may be capable of catalysing the development of NASH [54], though humanized features of metabolic disturbances will be absent in this model [11,54].

The American lifestyle induced obesity syndrome (ALIOS) diet is also a frequently used diet, in which high fat is combined with fructose-containing drinking water [56]. Compared to high trans-fat diets without additional fructose, these mice showed increased body weight and reduced insulin sensitivity, whereas no alterations in the degree of steatosis or liver transaminase levels were observed [11,56]. Moreover, in response to ALIOS diet, some pro-fibrogenic genes were found to be increased, while fibrosis was not detectable [56]. However, when mice were additionally administered a low weekly dose of intraperitoneal carbon tetrachloride (CCl_4_), these animals were shown to develop progressive stages of human fatty liver disease, ranging from simple steatosis to inflammation, fibrosis, and cancer [57]. Nevertheless, one important limitation of ALIOS is related to its dietary composition, as the amount of trans-fat per kilogram is greater than in commonly used fast foods [56].

Another promising model is the so-called diet-induced animal model of non-alcoholic fatty liver disease (DIAMOND). This model is based on wild-type C57BL/6 mice that were crossed with S129S1/svlmJ, a commonly used model to create mice with targeted mutations. After approximately four months of Western-type diet (42% kcal fat, 0.1% cholesterol, 3.1 g/L d-fructose, 18.9 g/L d-glucose), these mice have shown to recapitulate key physiological and metabolic features of human NASH [58]. However, a limitation of this dietary intervention is the high frequency of HCC development and suppression of cholesterol synthesis, which is substantially different from the human situation [12,58]. 

While in response to a chow diet, MS-NASH mice [59] (formerly known as FATZO/Pco mice, a cross between wild-type C57BL/6J and obesity-prone AKR/J mice [60]) spontaneously develop obesity [61], feeding these mice a Western-type fructose-supplemented diet resulted in progressive features of NAFLD/NASH [59,62]. Given the concomitant dysregulation in metabolic status, these data point towards a novel tool for studying NAFLD with high translational value. 

In summary, the above-described models have provided better insights into NAFLD/NASH pathogenesis. Nevertheless, it is noteworthy that these models failed to consistently achieve the full spectrum of human NASH, thereby limiting its preclinical validity.

### 2.2. Genetic Murine Models

Genetic animal models are essential for unravelling the underlying mechanisms related to the progression of NAFLD. Besides obesogenic *db/db* [45] and *ob/ob* mice [31,51], other models frequently used to study the total spectrum of human NAFLD and associated complications are based on genetically-modified mice in which the murine *ApoE* gene is being substituted by the human apolipoprotein E2 (APOE2) gene, referred to as the APOE2ki model [63]. Whereas wild-type C57BL/6 mice only developed simple steatosis in response to HFD, APOE2ki mice also displayed early-stage hepatic inflammation [63]. Yet, it is important to note that the inflammatory response did not persist in these mice [63]. Therefore, it is very likely that APOE2 gene is not the main gene responsible for the development of hepatic inflammation [25,63]. 

Relevantly, a complete lack of the murine *ApoE* gene, i.e., *ApoE^-/-^* model [64], resulted in hyperlipidemia after feeding these mice a high-fat diet [64]. Yet, under these conditions, mice spontaneously developed atherosclerotic plaques, while lacking humanized lipoprotein profiles [64], which suggests that this model is less suitable for human NAFLD research. 

Remarkably, existing knowledge on the low-density lipoprotein receptor (*Ldlr*), an important gene regulating the transport of non-modified lipids into macrophages, led to a major breakthrough in the field of NASH [25,63]. By a complete depletion of the *Ldlr*, mice fed a HFC diet for 3–12 weeks were able to resemble lifestyle-induced sustained hepatic inflammation [63]. Moreover, these mice displayed high levels of circulating LDL and low levels of HDL, thus closely mimicking the human lipoprotein profile [63]. Hence, this model is considered a physiological model to investigate early onset of NASH [25,63]. While the severity of fibrosis is rather mild, these mice have been shown to develop more fibrosis compared to regular C57BL/6 mice on a similar diet [25,63].

A more recent study investigated the relation between prostaglandin E2 and the severity of NASH, both in a clinical and preclinical context [65]. In general, prostaglandin E_2_, a member of the prostaglandin family, is known to play an important role during the inflammatory processes [66,67] in diseases such as rheumatoid arthritis and osteoarthritis [68]. However, its exact role in hepatic inflammation remains unknown. Henkel et al. [65] showed that a deficiency in the expression of enzymes responsible for murine prostaglandin E2 synthesis triggered a tumor necrosis factor α (TNFα)-dependent inflammatory response in the liver, thereby increasing the severity of diet-induced murine NASH. However, given that fibrosis and genotype-specific differences in macrophage infiltration were rather absent [65], it is very likely that the timing of feeding intervention (20 weeks) was not optimal to allow for advanced-stages disease development.

Another well-known model is based on a knock-in of the Patatin-like phospholipase domain-containing 3 (PNPLA3) polymorphism [20], which was found to be present in approximately one-fifth of our population [69,70]. PNPLA3 is a functional enzyme with acyltransferase and/or lipase activity towards phospholipids and/or triglycerides and retinyl esters, respectively [71]. When mice, carrying a mutation at position 148 of the *Pnpla3* gene were fed a high-sucrose diet, animals displayed increased levels of triglycerides and fatty acids, resulting in increased hepatic steatosis [72]. Nevertheless, no significant changes in hepatic inflammatory gene expression or fibrosis were observed [72]. Furthermore, in response to HFD, the development of hepatic steatosis was absent [72]. These data point towards diet as a primary trigger for PNPLA3-polymorphism-associated hepatic steatosis [72], thereby not covering the full spectrum of NAFLD.

Similarly, hepatic knockdown of transmembrane 6 superfamily member 2 (*Tm6sf2*), a gene responsible for regulating hepatic lipid metabolism and associated with increased susceptibility to human NAFLD [73], resulted in increased hepatic fat content and decreased VLDL secretion [74]. Though the specific role of *Tm6sf2* gene is not yet known, these data point towards its contribution to NAFLD development, and hence, its translational applicability. Remarkably, a recent meta-analysis showed that rs58542926 polymorphism significantly associated with chronic liver disease in the overall population [73]. These novel data pointed towards the diagnostic ability of TM6SF2-polymorphism to identify individuals at higher risk for developing NAFLD, cirrhosis, and HCC, as well as alcohol-dependent liver disease [73].

Another recent study investigated the role of Gankyrin (Gank) [75], an oncogene frequently expressed in several types of cancer [76] and a strong driver of liver proliferation. Using mice carrying a liver-specific deletion in Gank, it was shown that feeding a HFD for 6–7 months prevented fibrosis development in *Gank^-/-^* mice compared to HFD-fed wild-type mice [75]. While *Gank^-/-^* mice showed a higher degree of hepatic steatosis compared to HFD-fed wild-type mice, it has been postulated that hepatic steatosis protects the liver from fibrosis, and therefore liver proliferation could be a trigger for hepatic fibrosis [75]. Hence, the therapeutic potential of inhibiting hepatic proliferation as a strategy against NAFLD should be further investigated.

A more recent genetically-modified model that has become popular in NAFLD research is the obese *foz/foz* mouse model, which carries an 11-base pair truncating mutation in the Alström gene *Alms1* [12,14,77]. *Alms1* is widely expressed and disrupted by mutations in a human obesity syndrome, referred to as Alstroöm syndrome [78]. When feeding a HFD within a time frame of ten months, *foz/foz* mice displayed features of MetS, including obesity, hyperglycemia, hyperlipidemia, and insulin resistance [77]. In addition, these mice spontaneously developed steatosis, hepatic inflammation, and fibrosis [77]. Yet, while all *foz/foz* models have shown to develop obesity, some develop higher NAFLD activity scores and/or fibrosis than others, implying that the severity of NASH in these mice is inconsistent [14]. Moreover, given that the exact role of *Alms1* is not yet completely understood, the translational character of this model is rather limited. 

Another mouse model that spontaneously develops hepatic inflammation with rather a mild degree of fibrosis is the lean polygenetic fatty liver Shionogi (FLS) [79,80]. Remarkably, when backcrossing these mice with *ob/ob* mice, severe liver steatosis, inflammation, advanced fibrosis, and spontaneous HCC appeared to develop [81]. Nevertheless, due to its uncontrollable heterogeneity in disease onset, these models are scarcely used [82]. 

Other studies have used the hepatocyte-specific phosphatase and tensin homolog (PTEN)-deficient mouse model as a model for NAFLD [83,84]. PTEN, which is a phosphatase with activities towards both protein and lipids, was first discovered as a tumor suppressor protein [85]. More recently, its function as a metabolic regulator, also in the liver, has received increasing attention [85]. Indeed, PTEN-deficient mice were shown to display human-like lipid accumulation followed by liver fibrosis and HCC [83,84]. Nevertheless, these mice do not exhibit obvious human-like NASH features, such as increased circulating fatty acid levels and obesity, thereby limiting its translational potential. 

In contrast, others studied the role of augmenter of liver regeneration (*Alr*) in the context of NAFLD [86]. ALR, encoded by Growth Factor ERV1 homolog of *Saccharomyces cerevisiae* (*Gfer*), is an ubiquitous and multifunctional protein [86] that plays a vital role in liver generation, via regulating Natural Killer cell function [87], as well as other liver-related functions [88], including Kupffer cell activation [89]. Though mice deficient for *Alr* were prone to develop excessive hepatic steatosis [86], hepatic lipid accumulation was reversed at 4–8 weeks. Despite reversal of steatosis, mice developed hepatic inflammation, including hepatocellular necrosis, ductal proliferation, and fibrosis, which preceded dysplasia and HCC tumor development by nearly 60% one year after birth [11,86]. Hence, this model could aid in better understanding the progression from hepatic necrosis, inflammation, and fibrosis to carcinogenesis.

Another example is the melanocortin 4 receptor knockout (*Mc4r-/-*) mouse model [90]. MC4R is a G protein-coupled receptor expressed in hypothalamic nuclei being involved in regulating food intake and body weight [90]. Whereas chow-fed *Mc4r-/-* mice were shown to develop late onset obesity, hyperphagia, and simple steatosis due to genetic mutation, feeding a HFD induced ballooning degeneration, hepatic inflammation, and pericellular fibrosis [9]. In line with these results, using MRI-based techniques, Yamada et al. [91] recently showed that *Mc4r-/-* mice fed a HFD for 20 weeks developed obesity and NASH with clear signs of moderate fibrosis. Given their ability to functionally mimic the human NASH disease state, this model holds potential for studying hepatic dysfunction during advanced stages of NASH.

Alternative genetic models to study NASH progression and (spontaneously developing) HCC are the Tsumura-Suzuki Obese Diabetes (TSOD) mice, keratin 18-, NF-κB essential modulator (NEMO)-, and methionine adenosyltransferase 1A (MAT1a)-deficient models [92]. TSOD mice spontaneously developed NAFLD-related features, including T2D, obesity, glucosuria, hyperglycemia, and hyperinsulinemia without any special treatment [93]. Keratin 18 deficiency in mice serves as a model of NASH-associated liver carcinogenesis [94]. Liver-specific deletion of *NEMO* triggered steatosis, NASH, inflammatory fibrosis and subsequently HCC [95]. *Mat1a* gene deletion in mice impaired VLDL synthesis and plasma lipid homeostasis, thereby contributing to NAFLD development [96]. Yet, these models are generally less common and therefore less well-described in literature.

### 2.3. Chemically-induced Murine Models

As earlier described, alternative ways to explore the progression and/or regression of liver fibrosis and subsequent development of cirrhosis is by targeting the liver with CCL_4_ [57] or other chemotoxins, such as thioacetamide (TAA) [8,11,12]. For instance, biweekly administration of CCl_4_ for six weeks led to increased circulating aminotransferase and alkaline phosphatase levels in Balb/C mice [97]. In addition, CCL_4_ caused a dose-dependent progression of liver fibrosis [97]. However, the exact pathophysiological mechanism underlying hepatic fibrogenesis, in particular the role of hepatic stellate cells, requires further investigation. 

More recently, co-administration of TAA and western-type diet for eight weeks in wild-type C57BL/6 mice was shown to induce hepatic inflammation, severe diffuse fibrosis, and collagen deposition [98]. Nevertheless, due to significant reductions in body weight, these models do not optimally resemble humanized NASH etiology. 

One prominent model developed to better understand the progression from NAFLD to HCC is the STAM model, in which neonates received a low dose of streptozotocin, followed by a HFD starting from four weeks of age [99]. At ~6 weeks, ~8–12 weeks, and ~16–20 weeks of age, these mice developed inflammation and hepatocellular ballooning, progressive fibrosis, and HCC, respectively [99]. Concomitantly, these mice had reduced body weight and insulin levels compared to HFD-fed mice [99]. These data imply that NAFLD progression is likely an artificial process that does not accurately reflect human disease pathology, thereby limiting its preclinical potential.

Similar to the clinical situation, many preclinical NAFLD studies in dietary and genetic models demonstrated increased severity in males [100,101]. However, it should be noted that sex differences may vary between models and genotypes [25,102]. For instance, we previously showed that female *Ldlr*^−/−^ and APOE2ki mice fed HFD displayed a very early hepatic inflammatory response [25]. Similarly, it was demonstrated that female C57BL/6 wild-type mice fed a high-fructose diet developed greater hepatic inflammation despite having similar liver steatosis as compared to male mice [103]. Other studies showed that female juvenile NAFLD/NASH models displayed hepatic oxidative stress, whereas male animals rather developed hepatic inflammation [104]. In line with these results, it was more recently shown that high fat intake (60% kcal and 34.9% g fat, 20% kcal and 26.2% g protein, and 20% kcal and 26.3% g carbohydrate) by juvenile female mice contributed to NAFLD development, whereas similar fat intake by maternal-offspring (i.e., high-fat intake two weeks before conception and during gestation and lactation) resulted in the successful establishment of NASH [105]. These data suggest that maternal exposure, as well as the HFD component, contribute to the degree of NAFLD disease severity in juvenile female offspring [105].

### 2.4. Other Murine Models

Besides a role for genetic and dietary factors in preclinical NAFLD development, recent focus has also discretely shifted towards the relevance of housing conditions, thereby introducing a novel concept of thermoneutral housing (30–32 °C) [106]. Compared to standard housing conditions, mice housed under thermoneutral conditions were not only shown to induce a pro-inflammatory immune response, but also to deteriorate HFD-induced NASH progression [106]. Additionally, mice displayed increased intestinal permeability and alterations in gut microbiome, features mimicking the human situation [106]. Although these hallmarks could also partially refute the sex bias that is often observed in murine models of NAFLD, there were no signs of hepatic fibrosis, neither in male nor in female C57BL/6 wild-type mice [106]. Altogether, these data propose that a dietary stimulus is prerequisite for liver fibrosis development.

It is well-known that the liver is a central metabolic organ, whose functions are capable of adapting to rhythmical changes of environment. Indeed, it has been previously shown that circadian rhythm is driving oscillations in hepatic triglyceride levels, inflammation, oxidative stress, mitochondrial dysfunction, and hepatic insulin resistance [107,108]. Moreover, it has been recently suggested that chronic disruption of circadian rhythm may spontaneously induce the progression from NAFLD to NASH, fibrosis, and HCC [20,109], similar to the human situation, pointing towards its translational value.

Last but not least, there has also been increased awareness on the validity and reproducibility of preclinical studies on NAFLD [110,111,112]. For instance, it has been shown that murine liver fibrosis is affected by sampling variation [8]. More recently, Jensen et al. [2] demonstrated that feeding wild-type C57BL/6 mice a high-fat/high-fructose/high-cholesterol diet (40%/20%/2%) for 16 weeks resulted in significant intraindividual differences in fibrosis score and several hepatic biomarkers. Nevertheless, differences in sample variation were absent in other routinely used NAFLD rodent models [2]. These data pointed towards the importance of standardizing sampling site location during preclinical liver biopsy procedures, thereby supporting the ability to compare experimental outcomes between individual murine NASH studies.

### 2.5. Rat NAFLD Models 

In addition to the importance of murine models, preclinical studies on NAFLD pathogenesis are also frequently performed using rats. Rat models are thought to be more susceptible to HFD, and thus may display more severe and/or earlier histological features of NAFLD compared to mice [113]. A small selection of rat studies on NAFLD will be highlighted in this section.

Similar to mice, commonly used rat models refer to nutritional, genetic, and combined models (extensively reviewed elsewhere [114,115]), of which Sprague–Dawley [116,117], Wistar [118], and/or diabetic Zucker rats (fa/fa) [119] are well-known examples. For instance, Lieber et al. [120] and others [117] reported that Sprague–Dawley rats on a HFD (71% fat/11% carbohydrates/18% proteins) were able to develop insulin resistance, mild-to-marked steatosis, inflammation and/or fibrogenesis, thereby reproducing key features of human NASH. Yet, when fed a standard Lieber–DeCarli diet (35% fat, 47% carbohydrates, 18% proteins), rats displayed no signs of steatosis, inflammation, or fibrosis [120]. Diabetic Zucker rats, a well-characterized model of NAFLD, displayed similar features as its murine counterparts *ob/ob* and *db/db* mice, i.e., spontaneous development of severe obesity, steatosis, and insulin resistance [11]. Moreover, it was shown that Zucker rats are in need of an additional stimulus for onset of NASH [119]. Relevantly, when comparing 4 weeks of MCD diet between different rat models (i.e., Wistar, Long–Evans, and Sprague–Dawley rats), the Wistar strain was associated with the highest degree of hepatic fat accumulation [121], pointing either towards strain-dependency or the impact of dietary exposure time. 

Altogether, rat models are useful tools for providing additional valuable insights into the complex pathogenesis of steatosis/NASH (but not HCC), even though dietary or chemical interventions in these animals do not fully resemble the human situation.

## 3. Therapeutic Approaches in Preclinical NAFLD Models

In addition to exploring NAFLD etiology, preclinical models are critically important for testing how potential therapeutic drugs can interfere with the progression of this chronic disease [115].

Previously, Zheng et al. [27] chronically exposed HFC-treated mice with Ezetimibe, which is known to reduce plasma LDL by selectively binding to the intestinal cholesterol transporter Niemann–Pick type C1-like 1. After four weeks of Ezetimibe, significant improvements in fatty liver were observed, which were associated with a decrease in hepatic triglycerides, cholesteryl esters, and free cholesterol [27]. Additionally, chronic treatment with Ezetimibe resulted in significant reductions in plasma ALT activity, pointing towards its ability to serve as a novel treatment for HFC-induced NAFLD [27].

Trevaskis et al. [51] treated HFD-induced wild-type C57BL/6 and *ob/ob* mice with GLP-1R agonist AC3174, an exenatide analog. AC3174 treatment significantly reduced intrahepatic lipid accumulation, plasma triglycerides, and ALT levels, likely due to its contribution in weight loss [51]. Additionally, data suggested that AC3174 modestly improved the histological severity of fibrosis, which was demonstrated by a decrease in liver collagen-1 protein. Altogether, these findings suggest that AC3174 may play a beneficial role in the treatment of key aspects of fibrotic NASH.

Domitrovic et al. [97] investigated the therapeutic effect of luteolin in the context of liver fibrosis. Luteolin is a member of the flavonoid family, which has shown to exhibit hepatoprotective activity in acute liver damage, amongst others [122]. Administration of luteolin to CCL_4_-treated mice resulted in a dose-dependent reduction in hepatic fibrosis [97]. Although studies on the impact of luteolin in a more chronic model of liver fibrosis are desired, these data pointed towards therapeutic application of this drug in patients with hepatic fibrosis [97]. Similarly, in a study of Ganbold et al. [3], it was recently shown that administration of isorhamnetin, another natural flavonoid to human-like NASH mice, resulted in improved steatosis, liver injury, and fibrosis, pointing towards its therapeutic potential in NASH.

More recently, Khurana et al. [117] studied the role of inhibiting extracellular cathepsin D, a lysosomal enzyme that plays a role in lipid-related disorders, including NAFLD [123,124]. Using HFD-fed Sprague–Dawley rats, it was shown that inhibition of extracellular cathepsin D improved hepatic steatosis and reduced plasma levels of insulin and hepatic transaminases [117]. These data suggest that modulation of extracellular cathepsins may serve as a novel therapeutic modality for NAFLD [117].

Gehrke et al. [125] recently investigated eight to ten week-old wild-type C57BL/6 male mice that were fed an obesogenic diet (fructose/glucose supplementation in drinking water). In this study, mice were either challenged with voluntary wheel running or were kept on a sedentary lifestyle intervention [125]. Similar to well-known forced exercise models [126], voluntary wheel running protected these mice from HFD-induced pro-inflammatory and pro-fibrogenic states, as shown by decreased hepatic macrophage infiltration and improved fatty acid and glucose homeostasis. These data were in line with Kawanishi et al. [126], showing that exercise training reduced macrophage infiltration and adipose tissue inflammation by attenuating neutrophil infiltration in HFD-fed C57BL/6 mice. Thus, it is very likely that physical exercise exhibits beneficial effects and compensates for shortcomings of certain therapeutic approaches [125,126].

## 4. Clinical Relevance: Comparisons with Clinical Data

In general, NAFLD is considered the hepatic manifestation of MetS. Consequently, well-established therapeutic compounds against T2D and impaired lipid metabolism are thought to exert beneficial effects that mitigate the pathological features of NASH. One such example is the nuclear receptor peroxisome proliferator-activated receptor (PPAR) (extensively reviewed elsewhere [127]), due to its involvement in regulating lipid metabolism and inflammation. For instance, it was previously shown that hepatic *Pparα* inversely correlated with insulin resistance and NASH severity [128]. Remarkably, in this clinical study, histological improvements were positively associated with *Pparα* expression in patients with NASH [128], pointing towards the therapeutic potential of PPARα-agonists. Fibrates, which activate PPARα, are the most effective class of agents for lowering elevated triglyceride-rich lipoproteins [129]. While fenofibrate did not ameliorate liver histology in biopsy-proven NAFLD, a selective PPARα agonist, also known as Pemafibrate, did improve liver function in patients with dyslipidemia [130], which was previously established in diet-induced murine models of NAFLD [131].

Another isoform of PPAR, PPAR-δ, is involved in dyslipidemia and activation of PPAR-δ by means of a selective agonist, referred to as seladelpar (MBX-8025), beneficially affected plasma lipid levels and showed favorable trends in insulin resistance and waist circumference in patients with dyslipidemia [132]. More recently, these data were further supported using in-vivo studies, showing that seladelpar improved glucose metabolism, as well as plasma and hepatic lipid levels in obese *foz/foz* mice [133]. Collectively, these data pointed towards seladelpar as a potential novel therapy for NASH.

Another strategy is based on insulin-sensing drugs, known as Thiazolidinediones, which target the PPAR-γ isoform [134]. Examples of PPAR-γ agonists are lobeglitazone, pioglitazone, and rosiglitazone, of which the latter two are currently off-label and off-market, respectively [127]. When diabetic patients with NAFLD were treated with lobeglitazone, patients showed improvements in hepatic steatosis, glycemic, and lipid profiles, as well as liver enzyme levels [134]. In line, it was shown that HFD-induced obese mice treated with lobeglitazone improved glucose homeostasis as well as hepatic and plasma lipid levels [135]. Hence, these data pointed towards lobeglitazone as a potential treatment option for NAFLD.

Relevantly, the use of elafibranor (dual PPAR-α/δ agonist, clinical phase III trial) as a single drug regime is thought to be promising with regard to NASH, as shown by significant improvements of human NASH pathology without deteriorating hepatic fibrosis [136]. These data were further corroborated by findings in rodent studies of NAFLD [137]. Additionally, the development of anti-fibrotic therapies has recently received increasing attention [138]. Therefore, due to its multifactorial character, current treatment modalities should focus on both the reversal of NASH [139] and fibrosis [138]. 

Strategies, which currently hold clinical potential in late-stage drug development, include specific complementary agonists, i.e., for PPAR receptor subtypes and farnesoid X receptor (FXR) [139]. Using AMLN diet-induced obese mice with biopsy-confirmed NASH, Roth et al. [139] demonstrated that combined treatment with Elafibrinor and obeticholic acid (FXR agonist) significantly ameliorated histological features of steatosis, inflammation and fibrosis. Additionally, compared to single regimens, combined treatment targeted hepatic molecular mechanisms, thereby further improving NASH and fibrogenesis [139].

To conclude, in addition to the selective—but relevant—approaches described above, additional therapeutic options for NAFLD have been studied in preclinical and/or clinical settings [16], or are currently under investigation.

## 5. Conclusions

Although current preclinical NAFLD models can be considered indispensable tools for studying chronic liver disease pathology, it should be noted that the majority of existing rodent models mainly focus on certain stages of the disease rather than the total spectrum. Additionally, it is noteworthy that the NAFLD disease progression greatly varies across different strains [140]. Therefore, depending on its research question, careful model selection is highly recommended. Such selection should also properly consider sex, age, and hormonal status and must be based on prior knowledge, as it will have a large impact on data interpretation and its translational potential. Thus, our common goal is to establish an ideal preclinical model that—in addition to developing hepatic inflammation and fibrosis, along with obesity, high cholesterol, and insulin resistance—also responds to promising therapeutic interventions. This implies that future studies should continue focusing on recapitulating the multifactorial character of human NAFLD in preclinical models. 

## Figures and Tables

**Figure 1 biomedicines-08-00028-f001:**
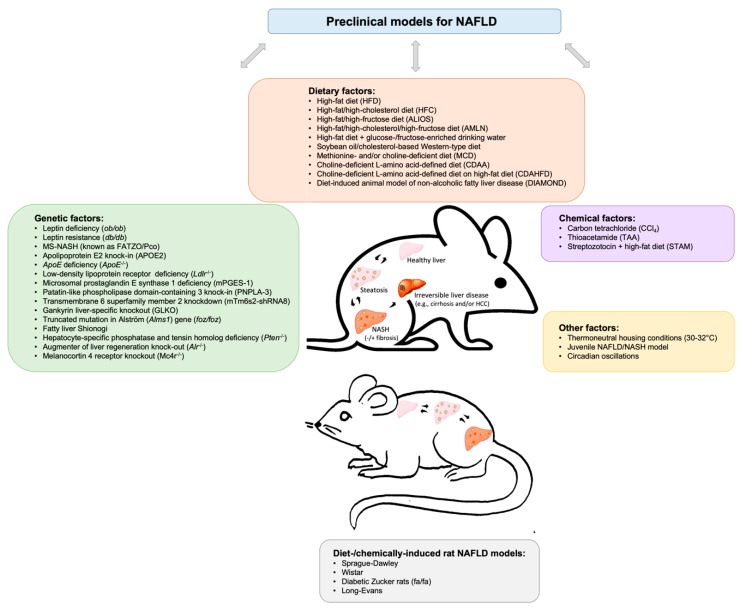
Brief schematic overview of existing preclinical models involving a variety of dietary, genetic, chemical, and other factors to study non-alcoholic fatty liver disease (NAFLD).

**Table 1 biomedicines-08-00028-t001:** Overview of commonly used dietary, genetic, chemically-induced, and other murine models of non-alcoholic steatohepatitis (NASH).

Type of Mouse Model	Treatment or Intervention	Phenotypical Outcome and Relevance to Human Disease	Author(s)
**Dietary Models**			
High-fat diet (HFD)	HFD containing 60% fat administered to C57BL6/J	After 10-12w, induction of obesity, insulin resistance and hyperlipidemia. After long-term exposure (36w), no or only minimal signs of inflammation and fibrosis. After chronic feeding (80w), hepatic steatosis, cell injury, portal & lobular inflammation and fibrosis.	Velázquez et al. [17], Ito et al. [19], Chen et al. [20], Vonghia et al. [21]
Atherogenic diet	Diet containing 1.25% cholesterol and 0.5% cholate	Increased plasma and liver lipid levels. From 6–24 weeks, induction NASH with hepatocellular ballooning in a time-dependent manner.	Matsuzawa et al. [24]
High-fat atherogenic diet	HFD containing 1.25% cholesterol and 0.5% cholate	Exacerbated NASH features including hepatic insulin resistance, oxidative stress, activation of hepatic stellate cells.	Matsuzawa et al. [24], Montandon et al. [41], Larter et al. [42]
High-fat/high-cholesterol diet (HFC)	HFC containing 21% milk butter, 0.2% cholesterol	After short-term HFC diet, only steatosis in C57BL/6 mice. Steatosis with severe inflammation in female *Ldlr*^-/-^ and APOE2ki hyperlipidemic mice. After seven days, severe hepatic inflammation but no steatosis in male hyperlipidemic mice. After seven months, development of obesity, hepatomegaly, hepatic steatosis and varying degrees of steatohepatitis in C57BL/6 mice.	Wouters et al. [25,26] Zheng et al. [27]
High-fat/high-cholesterol/ high-fructose diet (AMLN)	Diet containing 40% high-fat and 22% fructose, supplemented with ~18% trans-fat and 2% cholesterol	After 26–30w, marked steatosis, moderate lobular inflammation and hepatocellular ballooning in C57BL/6 and *ob/ob* mice.	Clapper et al. [28], Kristiansen et al. [29]
Gubra amylin diet (GAN)	High-fat (40 kcal-%, of which 0% trans-fat and 46% saturated fatty acids by weight), fructose (22%), sucrose (10%), cholesterol (2%)	After 8-16w, more pronounced weight gain and a highly similar phenotype of biopsy-confirmed fibrotic NASH in C57BL/6 and *ob/ob* mice.	Boland et al. [33]
High-fat/high-fructose/high-cholesterol	Composed of 41% fat, 30% fructose, 2% cholesterol	Induction NASH in various models.	Abe et al. [34], Kennedy et al. [35], Suto et al. [36]
Soybean-oil-based Western-type diet	Western-type diet containing 25g/100 g n-6-PUFA-rich soybean oil +/- 0.75% cholesterol	After long-term exposure (20w), hepatic steatosis, inflammation and fibrosis, weight gain, insulin resistance, hepatic lipid peroxidation and oxidative stress in C57BL/6 mice.	Henkel et al. [37]
High-caloric cholesterol-free HFD	Composed of lard (21g/100g)/soy-bean oil (3g/100g)/5% fructose in drinking water	Only mild steatosis. No signs of hepatic inflammation and fibrosis.	Wouters et al. [25,26], Henkel et al. [37], Subramanian et al. [38], Mari et al. [39], Savard et al. [40]
Choline-deficient diet	C57BL/6 mice were fed HFD (45% of calories) for 8 weeks. During the final 4 weeks, diets were choline-deficient (or choline-supplemented)	Amplified liver fat accumulation, while improved glucose tolerance.	Raubenheimer et al. [48]
Methionine/choline-deficient diet (MCD)	Diet lacking methionine and choline, but containing high sucrose (40%) and moderate fat (10%)	After 2w, severe steatohepatitis with elevated serum AST and ALT levels. After 10w, additional Kupffer cell infiltration and irreversible fibrosis. After 1.5-4w, no signs of insulin resistance.	Santhekadur et al. [12], Montandon et al. [41], Itagaki et al. [47], Rinella et al. [52], Al Rajabi et al. [53]
Choline-deficient L-amino acid-defined diet (CDAA)	Choline-deficient L-amino acid-defined diet containing carbohydrates (68,5%), proteins (17,4%) and fats (14%)	Within a few weeks, fatty liver followed by mild features of NASH in C57BL/6J mice. After >20w, mild-to-moderate fibrosis and insulin resistance.	Van Herck et al. [11], Matsumoto et al. [54], Miura et al. [55]
Choline-deficient L-amino acid-defined diet on high-fat diet (CDAHFD)	Choline-deficient, L-amino acid-defined, HFD consisting of 60 kcal% fat and 0.1% methionine by weight	Excessive liver fat accumulation, increased circulating liver enzymes and progressive hepatic fibrosis.	Matsumoto et al. [54]
High-fat/high-fructose diet (ALIOS)	HFD with fructose-containing drinking water. Additional administration of a low weekly dose of intraperitoneal carbon tetrachloride (CCl_4_)	After 16w, substantial steatosis with necro-inflammatory changes and increased ALT levels. No difference in steatosis degree or ALT levels if compared to without additional fructose. Development of progressive stages of human-like fatty liver disease.	Tetri et al. [56], Tsuchida et al. [57]
Diet-induced animal model of non-alcoholic fatty liver disease (DIAMOND)	High fat/carbohydrate diet (Western diet) with 42% kcal from fat, containing cholesterol (0.1%), with a high fructose/glucose solution (23.1 g/L d-fructose +18.9 g/L d-glucose)	After 16w, obesity, liver injury, dyslipidemia and insulin resistance, sustained up to 52w. Parent strains 129S1/SvImJ or C57BL/6 lacked insulin resistance and steatohepatitis or developed delayed insulin resistance.	Santhekadur et al. [12], Asgharpour et al. [58]
High-fat diet + glucose/ fructose-enriched drinking water	Obesogenic diet containing ((35.5% w/w) crude fat (58 kJ%), 22.8 MJ/kg = 5.45 kcal/g) and fructose (55% w/v) and glucose (45% w/v) enriched drinking water. After 8 weeks of dietary feeding, mice were randomly assigned to a voluntary wheel running group or a sedentary group.	Voluntary wheel running prevented HFD-induced pro-inflammatory / fibrogenic states in C57BL/6 mice. Hepatic steatosis was prevented by alterations in key liver metabolic processes.	Gehrke et al. [125]
**Genetic Models**			
Leptin deficiency (*ob/ob*)	Leptin-deficient (*ob/ob*) mice are predisposed to develop NASH and fibrosis, whereas not when maintained on regular chow diet. Treatment with high-fat/high-fructose/high-cholesterol diet.	Lack the ability to spontaneously develop hepatic inflammation. After 12-26w, increased adiposity, total cholesterol and elevated plasma liver enzymes upon diet high in trans-fat (40%), fructose (22%) and cholesterol (2%). After treatment with high-fat/high-fructose/high-cholesterol diet, development of metabolic, histologic and transcriptomic features similar to human NASH.	Kristiansen et al. [29], Abe et al. [34], Trevaskis et al. [51]
Leptin resistance (*db/db*)	*db/db* mice are deficient in the leptin receptor, with dramatic elevations in circulating leptin concentrations. Dietary intervention with an MCD diet for 4 weeks.	Lack the ability to spontaneously develop hepatic inflammation and thus needs to be combined with a nutritional model for NASH. After 4w MCD diet, mice displayed marked hepatic inflammation and fibrosis.	Kennedy et al. [35], Sahai et al. [45], Hummel et al. [50]
MS-NASH (FATZO/Pco)	Mice spontaneous development of obesity	After 20w of fructose-supplemented diet, hepatic steatosis, lobular inflammation, ballooning and fibrosis.	Sun et al. [62]
Apolipoprotein E2 knock-in (APOE2)	Murine ApoE replaced by the human APOE2 gene	After 12w of HFC, steatosis in conjunction with early but not sustained hepatic inflammation.	Wouters et al. [25,26], Bieghs et al. [63]
ApoE deficiency (*ApoE*^-/-^)	Complete deficiency in the murine ApoE gene	After 7w of Western diet, abnormal glucose tolerance, hepatomegaly, weight gain and full spectrum of NASH, while lacking humanized lipoprotein profiles.	Schierwagen et al. [64]
Low-density lipoprotein receptor deficiency (*Ldlr*^-/-^)	Complete deficiency of the murine Ldl receptor, an important gene regulating the transport of non-modified lipids into macrophages	After 3-12w of HFC diet, resemblance to lifestyle-induced early-onset hepatic inflammation. High and low levels of circulating LDL and HDL, respectively, closely mimicked the human lipoprotein profile. Development of mild fibrosis.	Wouters et al. [25,26], Bieghs et al. [63]
Microsomal prostaglandin E synthase 1 (mPGES1) deficiency	Mice with global deletion of mPGES-133 were backcrossed on C57BL/6J	TNFα-dependent inflammatory response in murine liver. Increased severity of diet-induced murine NASH.	Henkel et al. [65]
Patatin-like phospholipase domain-containing 3 (PNPLA-3) knock-in	Mice carried I148M mutation in the Pnpla3 gene and were fed a high-sucrose or HFD diet for 4 weeks	Accumulation of PNPLA3 on lipid droplets. Development of hepatic steatosis.	Smagris et al. [72]
Transmembrane 6 superfamily member 2 knockdown (mTm6s2-shRNA8)	Adeno-associated virus-mediated short hairpin RNA knockdown of Tm6sf2 in liver of C57BL/6J mice	Increased hepatic fat content and decreased VLDL secretion, recapitulating the effects observed in humans carrying the TM6SF2-167Lys mutation	Kozlitina et al. [74]
Gankyrin liver-specific knockout (GLKO)	Cre-Alb mice were backcrossed with LoxP-Gank mice	Gankyrin generally drives liver proliferation. After 6-7 months of HFD, higher degree of hepatic steatosis but prevention of fibrosis development in GLKO mice compared to wild-type mice.	Cast et al. [75]
Truncated mutation in Alström (Alms1) gene (*foz/foz*)	11-base pair truncating mutation in the Alström gene ALMS1. Lack of knowledge regarding the exact role of Alms1	After 6 months of HFD, MetS features, including obesity, hyperglycemia / lipidemia and insulin resistance. Mice spontaneously develop steatosis, hepatic inflammation and fibrosis.	Santhekadur et al. [12], Jiang et al. [14], Arsov et al. [77]
Fatty liver Shionogi	Spontaneous development of hepatic inflammation with rather a mild degree of fibrosis. Uncontrollable heterogeneity in disease onset	Backcrossing with *ob/ob* mice resulted in severe liver steatosis, inflammation, advanced fibrosis and spontaneous HCC	He et al. [82]
Hepatocyte-specific phosphatase and tensin homolog deficiency (*Pten*^-/-^)	PTEN deficiency specific in the liver	After 40w of age, steatosis, inflammation and fibrosis in the liver. After 74-78w of age, HCC was present in 83% of males and 50% of female mice.	Watanabe et al. [83], Takakura et al. [84]
Augmenter of liver regeneration knock-out (*Alr*^-/-^)	Liver-specific deletion of augmenter of liver regeneration	4-8w after birth, steatohepatitis with hepatocellular necrosis, ductular proliferation and fibrosis. 1y after birth, HCC in nearly 60% of the mice	Van Herck et al. [11], Gandhi et al. [89]
Melanocortin 4 receptor knockout (*Mc4r*^-/-^)	Mice with targeted disruption of melanocortin 4 receptor, which is a seven-transmembrane G protein–coupled receptor that is expressed in the hypothalamic nuclei	Development of simple steatosis. Upon feeding HFD, development of human-like NASH, including obesity, insulin resistance and dyslipidemia. After 20w HFD, obesity and NASH with clear signs of moderate fibrosis, functionally mimicking the human NASH disease state.	Itoh et al. [90], Yamada et al. [91]
**Chemically-induced Models**			
Carbon tetrachloride (CCL_4_)	Biweekly injections of CCl_4_	After 6w, increased circulating liver enzymes and dose-dependent progression of liver fibrosis in Balb/C mice	Domitrovic et al. [97]
Thioacetamide (TAA)	Three times/week IP injection thioacetamide (75mg/kg) in combination with western-type diet	After 8w, hepatic inflammation, severe diffuse fibrosis and collagen deposition in C57BL/6 mice	Hansen et al. [8] Van Herck et al. [11] Santhekadur et al. [12]
Streptozotocin + high-fat diet (STAM)	200 μg streptozotocin at 2 days after birth and feeding ad libitum with high-fat diet at 4 weeks of age	Between 6-20w of age, hepatic inflammation, hepatocellular ballooning, progressive fibrosis and HCC. Reduced body weight and insulin levels compared to HFD-fed mice	Fujii et al. [99]
**Other models**			
C57BL/6 background	Mice were housed in separate specific pathogen-free units maintained at either 22°C (standard) or 30-33°C (thermoneutral)	After 24w of thermoneutral housing, exacerbated HFD-driven NAFLD pathogenesis. Increased intestinal permeability and alterations in gut microbiome, mimicking the human situation.	Giles et al. [106]
C57BL/6 background	Juvenile NASH model: immediately after weaning, mice were fed HFC diet for a total of 16 weeks (4, 8, 12 and 16 weeks of diet)	Hepatic oxidative stress in female juvenile NAFLD/NASH models, whereas hepatic inflammation in males	Marin et al. [104]
C57BL/6 background	Juvenile NAFLD/NASH model: HFD were administered 2 weeks before conception and during gestation and lactation	Offspring HFC intake resulted in NAFLD, maternal-offspring fat intake contributed to NASH in juvenile female mice	Zhou et al. [105]
Models with circadian oscillations (e.g. *Per1/2*^-/-^ or liver-specific *Bmal1* knockout mice)	The effects of feeding time and circadian clocks on murine liver	Circadian rhythm drives oscillations in hepatic triglyceride levels, inflammation, oxidative stress, mitochondrial dysfunction and hepatic insulin resistance. Chronic disruption of circadian rhythm may spontaneously induce the progression from NAFLD to NASH, fibrosis and HCC	Adamovich et al. [107], Jacobi et al. [108], Kettner et al. [109]
C57BL/6 background	Mice (and other rodent models) were fed a high-fat/high-fructose/high-cholesterol for 16 weeks	Significant intraindividual differences in fibrosis score and hepatic biomarkers pointed towards the importance of standardizing sampling site location during preclinical liver biopsy procedures	Jensen et al. [2]

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
