# Peer review of "NAFLD Preclinical Models: More than a Handful, Less of a Concern?"

_biomedicines, 2020, doi:10.3390/biomedicines8020028_

Round 1
Reviewer 1 Report
The authors have done tremendous job compiling most of the models associated with NAFLD spectrum. The review could be further improved if the authors address few of the following things.
Under the diet based model section; it will be helpful to add atherogenic high fat diet and define the difference between regular HFD model MCD diet model is only discuss in the context of fibrosis however it is an important model for NASH associated inflammation aspect. It should be included under diet based model section rather than discussing it in fibrosis perspective.
The organization of the manuscript can be improved. It would be good to have sections titles that group different models. Besides start the section with what these model address and end with shortcomings that leads to next section/model.
The summary table is awesome ! but again the grouping can help users understand the things in more convenient and lucid way.
Author Response
Reviewer 1 comments:
We thank the reviewer for his/her kind words and useful feedback that improved the quality of our
manuscript. All changes have been highlighted in yellow.
1. Under the diet-based model section, it will be helpful to add atherogenic high fat diet and define the difference between regular HFD model.
According to the reviewer’s suggestion, we clarified the regular HFD model (Page 3, lines 6-7 and lines 7-17). Furthermore, we defined the difference between the HFD and atherogenic diet (Page 3, lines 24-28).
2. MCD diet model is only discussed in the context of fibrosis. However, it is an important model for NASH-associated inflammation aspect. It should be included under diet-based model section rather than discussing it in fibrosis perspective.
We have restructured the text (as indicated also in point 3) and included the MCD model under ‘Dietary murine models’ (Page 4, lines 25-52). Using a recent study by Montandon et al. (2019), we now elaborated on the MCD model in the context of NASH-associated inflammation (Page 4, lines 25-38).
3. The organization of the manuscript can be improved. It would be good to have section titles that group different models.
We thank the reviewer for his/her valuable remark. We restructured the manuscript and made a clear distinction by grouping different models, as indicated below:
- Section 2: Insights into available preclinical models for NAFLD (Page 2, line 14).
o Sub-section 2.1: Dietary murine models (Page 3, line 1 – Page 5, line 34)
o Sub-section 2.2: Genetic murine models (Page 5, line 35 – Page 7, line 42)
o Sub-section 2.3: Chemically-induced murine models (Page 7, line 43 – Page 8, line 25)
o Sub-section 2.4: Other murine models (Page 8, lines 26 – Page 9, line 2)
o Sub-section 2.5: Rat NAFLD models (Page 9, line 3 – Page 9, line 24)
4. Start the section with what the models address and end with shortcomings that leads to the next section/model.
Prior to clarifying the models, we now also included what the models address. In addition, throughout the text, we have included the shortcomings (or clinical potential) of the model, which we have highlighted in yellow.
5. The summary table is awesome! Grouping can help users understand the things in a more convenient and lucid way.
We thank the reviewer for the useful suggestion. According to the outline of the text, we have clustered all the information per section. Furthermore, we summarized the text to enlarge the table and increase its readability.
Reviewer 2 Report
Biomedicines – Revision
In the manuscript entitled “NAFLD mouse models: more than a handful, less of a concern?”, Oligshlaeger and Shiri-Sverdlov investigated the current literature regarding the most common mouse models used to study NAFLD and NASH. This review is of particular importance, especially given the great concern raised in the last years concerning NAFLD. Authors illustrated genetically-, dietary- and chemically-induced murine models employed to deciphering NAFLD pathophysiology, plus they considered other factors (such sex, housing conditions, circadian rhythms) that may affect the outcome of this disease in mice. However, in my opinion, this manuscript requires substantial improvements. Major concerns are listed below.
In the introduction section, authors should briefly discuss the current model for NAFLD/NASH onset and progression, given peculiar attention to the role of inflammation, since this last one is a recurrent feature of the murine models described later in the manuscript. Introduction: authors described cirrhosis and HCC as a continuum from NAFLD. However, several studies indicated that these two diseases could also onset without an established NAFLD/NASH condition. Please, specify it. Authors should separate the “2. Brief insights…” section in different subsections (i.e. Dietary factors; Genetic Factors; Chemical Factors; Other Factors). This will allow a more fluid reading by the reader and an easy access to the data provided. Page 2, line 51. Is the term “gnotobotic” used to indicate germ free mice? In any case, please specify, in order to make the test more accessible also to people that are not expert in the field. Page 2, line 53. Authors said “thereby mimicking features of human NASH”. Is this referred only to microbiota changes or it is also valid for liver phenotype? Please, be more precise and clearer. Page 2, lines 60-65. This paragraph is difficult to read, since it is not clear if the authors are speaking of one or two type of diets. Please, rewrite with proper indications. Page 2, line 64. I would prefer using the word “genotypes” and then refer to C57BL/6J as wild type, rather than “strains”. Ob/ob and KK-Ay are not strains! Page 2, lines 67-68 versus page 3, lines 104-106. These two statements are quite contradictory. Please, modify the test. Page 3, lines 74-76. Authors seems to claim to cholesterol supplementation for NASH onset in mice. Is it correct? If not, please rewrite this section. Page 4, lines 157. “However, in contrast to excessive weight increase, these models showed significant reductions in weight”. This sentence is contradictory. Please, correct it. Throughout the manuscript, authors usually referred to gene or genotype that are not so commonly used. Please, for each of these, spend few words describing gene/protein function and eventually the process in which they are involved. Table 1 is too small, therefore it was not possible to make an opportune evaluation.
Author Response
Reviewer 2 comments:
We would like to thank the reviewer for his/her insights and useful remarks and for acknowledging
the relevance of our work. All changes have been highlighted in yellow.
1. In the introduction section, authors should briefly discuss the current model for NAFLD/NASH onset and progression, given peculiar attention to the role of inflammation.
In section 1 ‘Introduction’, we have now underlined the inflammatory component of NASH (Page 1,
line 27-29). In addition, we elaborated on the role of inflammation in the context of an ‘ideal’ model
(Page 1, line 33-35). Furthermore, we shortly pointed towards currently available models that have
broadened our knowledge with regard to NAFLD progression, in particular (early-onset) low-grade
inflammation (Page 1, line 40 – Page 2 line 6), which are being discussed in-depth in section 2 ‘Insights
into available preclinical models for NAFLD’. All changes have been highlighted in yellow.
2. Introduction: authors described cirrhosis and HCC as a continuum from NAFLD. However, several studies indicated that these two diseases could also onset without an established NAFLD/NASH condition. Please, specify it.
We thank the reviewer for his/her useful suggestion. We have clarified the text by indicating that
NASH can increase the risk of cirrhosis and hepatocellular carcinoma. To further specify, we included
a sentence mentioning that, in contrast to alcoholic liver disease- and viral hepatitis-induced HCC,
NASH-related HCC is currently the most rapid growing indication for liver transplant in HCC patients
(Page 1, lines 28-32).
3. Authors should separate the former “2. Brief insights…” section in different subsections. This will allow a more fluid reading by the reader and an easy access to the data provided.
We thank the reviewer for his/her valuable remark. We restructured the manuscript and made a clear
distinction by grouping different models, as indicated below:
- Section 2: Insights into available preclinical models for NAFLD (Page 2, line 14).
o Sub-section 2.1: Dietary murine models (Page 3, line 1 – Page 5, line 34)
o Sub-section 2.2: Genetic murine models (Page 5, line 35 – Page 7, line 42)
o Sub-section 2.3: Chemically-induced murine models (Page 7, line 43 – Page 8, line 25)
o Sub-section 2.4: Other murine models (Page 8, lines 26 – Page 9, line 2)
o Sub-section 2.5: Rat NAFLD models (Page 9, line 3 – Page 9, line 24)
4. Is the term “gnotobotic” used to indicate germ free mice?
We have deleted the word ‘gnotobiotic’ and replaced it with the word ‘germ-free mice’ to avoid any
confusion (Page 3, lines 17-19).
5. Authors said “increases in the Firmicutes phylum, thereby mimicking features of human NASH” (Former page 2, line 53). Is this referred only to microbiota changes or it is also valid for liver phenotype? Please, be more precise and clearer.
We thank the reviewer for his/her useful remark. In the study of Velazquez et al. (2019), it was shown
that increased Firmicutes phyla pointed towards diet-induced gut-microbial dysbiosis, a well-known
microbial event that has been previously observed in NAFLD patients. We have now clarified this
accordingly (Page 3, lines 17-22).
6. This paragraph (former Page 2, lines 60-65) is difficult to read, since it is not clear if the authors are speaking of one or two type of diets. Please, rewrite with proper indications.
To improve the readability of the text, we have clarified it by splitting the paragraph. First, we discuss the effect of obesogenic trans-fat-free Gubra Amylin NASH (GAN) diet, which we compared to the AMLN diet (Page 3, line 45 – Page 4, line 3). In the second paragraph, we discuss an alternative fastfood-
like nutritional regime based on high-fat/high-fructose/high-cholesterol in different genotypes
(Page 4, lines 4-12). We backed-up the text with additional references.
7. I would prefer using the word “genotypes” and then refer to C57BL/6J as wild type, rather than “strains”. Ob/ob and KK-Ay are not strains!
We removed the word ‘strains’ from the text and replaced this word with ‘genotypes’ accordingly
(Page 4, line 5 and Page 8, line 14). Furthermore, throughout the entire text, we have added the word
‘wild-type’ when discussing C57BL/6 mice.
8. The statements formerly made in Page 2, lines 67-68 versus page 3, lines 104-106 are quite contradictory. Please, modify the text.
We thank the reviewer for his/her useful remark. We have clarified these sentences, as indicated
below:
‘Given that the metabolic, histologic and transcriptomic features observed in ob/ob mice were similar
to human NASH, this model may be further explored as a potential preclinical tool to discover novel
drugs for NASH [33]’ (Page 4, lines 10-12)
‘Further, though ob/ob and db/db models serve as useful preclinical tools that mimic insulin resistance
as observed in humans, it should be kept in mind that these mice bear mutations that are not prevalent
in obese humans or NASH patients.’ (Page 4, lines 50-52)
9. Authors seem to claim to cholesterol supplementation for NASH onset in mice. Is it correct?
We agree with the reviewer’s point. In section 2.1 ‘Dietary murine models’, we have now defined how
cholesterol supplementation can contribute to the onset of NASH (Page 4, lines 13-22).
10. Former Page 4, lines 157: “However, in contrast to excessive weight increase, these models showed significant reductions in weight”. This sentence is contradictory. Please, correct it.
We thank the reviewer for pointing towards this mistake. We have rewritten this sentence as indicated
below:
‘Nevertheless, it is noteworthy that all MCD models rather showed significant reductions in weight,
concomitant loss in liver mass, cachexia as well as low serum levels of insulin, fasting glucose, leptin
and triglycerides and a lack of insulin resistance [13,45,53]. Given that these preclinical observations
are opposite to the effects seen in overweight and obese individuals with NAFLD, these data suggest
that the use of MCD models as preclinical tools to represent human NAFLD is rather limited [54].’ (Page
4, lines 45-52).
11. Throughout the manuscript, authors usually referred to gene or genotype that are not so commonly used. Please, for each of these, spend few words describing gene/protein function and eventually the process in which they are involved.
We have now better explained less-common genotypes or genes/proteins. These changes have been
highlighted in yellow throughout the entire text.
12. Table 1 is too small, therefore it was not possible to make an opportune evaluation.
We thank the reviewer for his/her remark. We summarized the text, which allowed us to enlarge the
table, thereby increasing its readability.
Reviewer 3 Report
Non-alcoholic fatty liver disease (NAFLD) is a spectrum of liver diseases is currently a major concern for the public health and metabolic syndrome. There are a significant number of pre-clinical models used to stablished NAFLD diseases and therapeutic approaches have been used to treat NAFLD. The current review is mainly focused on summarizing all mice models available and their pros and cons. However, I have a few concerns about this.
The manuscript is written in very complex, long sentences and a number of places without any references. There are a number of important statements without any references. The manuscript needs a significant rewriting to make short sentences along with informative information and references. The authors aimed to summarize mouse models available for NAFLD, the section is well described. However, I am wondering why the authors did not consider RAT model?
I felt this is more suitable for a PhD thesis where a candidate needs to justify their “mouse model”. I do not see any significant contribution to this review in the existing literature.
In addition to the different animal models, the authors may also consider the therapeutic aspect or rescue NAFLD aspect for this review. In other words, the authors may consider adding a summary for rescuing NAFLD in animal models (pharmacological or genetic or feeding). Other than adding a significant section (as a suggestion, i. all available preclinical models; ii. Therapeutic approaches to treat NFLD in the animal; iii. Comparing with clinical data) that can help the existing literature, I do not see this review can contribute to the existing literature.
Author Response
Reviewer 3 comments:
We would like to thank the reviewer for his/her insights and useful remarks. All changes have been
highlighted in yellow.
1. The manuscript is written in very complex, long sentences
We have rewritten long sentences into shorter ones, which makes it easier to read the manuscript.
Moreover, to further increase readability of the text, we have introduced sub-sections:
- Section 2: Insights into available preclinical models for NAFLD (Page 2, line 14).
o Sub-section 2.1: Dietary murine models (Page 3, line 1 – Page 5, line 34)
o Sub-section 2.2: Genetic murine models (Page 5, line 35 – Page 7, line 42)
o Sub-section 2.3: Chemically-induced murine models (Page 7, line 43 – Page 8, line 25)
o Sub-section 2.4: Other murine models (Page 8, lines 26 – Page 9, line 2)
o Sub-section 2.5: Rat NAFLD models (Page 9, line 3 – Page 9, line 24)
2. There are a number of important statements without any references.
We thank the reviewer for his/her valuable point. To back-up important statements, we have inserted
the corresponding reference(s), as highlighted in yellow. Furthermore, we have added ~60 references
in order to back-up additional information based on this revision.
3. I am wondering why the authors did not consider RAT model
We thank the reviewer for his/her useful remark. Indeed, this Review mainly focuses on mouse
models. Nevertheless, to make this manuscript more complete, we have now included a paragraph
highlighting several rat models for NAFLD (Sub-section 2.5, Page 9, lines 3-24). As such, we have
changed the title of this review accordingly and implemented these models in Figure 1.
4. In addition to the different animal models, the authors may also consider the therapeutic aspect or rescue NAFLD aspect for this review. As a suggestion, i. all available preclinical models; ii. Therapeutic approaches to treat NFLD in the animal; iii. Comparing with clinical data) can help the existing literature.
We thank the reviewer for his/her valuable point. In addition to describing all available models
(section 2), we now added a paragraph, discussing therapeutic approaches in preclinical NAFLD
models (Section 3, Page 12, line 2 – Page 13, line 22). In addition, we added a fourth section, where
we address the results of important clinical trials and compare these to the findings in vivo (Section 4
‘Clinical relevance: comparisons with clinical data’, Page 13, line 23 – Page 14, line 17)
Round 2
Reviewer 2 Report
The authors extensively reviewed their manuscript. The new version of the manuscript is accurate and easily readable. No further changes are required.
Reviewer 3 Report
The manuscript reads well. It is hard to read the figure1 texts. The authors might need to consider making it bigger or showing it in a different way.